# Reviewing the WHO Tube Bioassay Methodology: Accurate Method Reporting and Numbers of Mosquitoes Are Key to Producing Robust Results

**DOI:** 10.3390/insects13060544

**Published:** 2022-06-14

**Authors:** Giorgio Praulins, Daniel P. McDermott, Angus Spiers, Rosemary Susan Lees

**Affiliations:** 1Innovation to Impact (I2I), Liverpool School of Tropical Medicine, Pembroke Place, Liverpool L3 5QA, UK; angus.spiers@innovation2impact.org; 2Department of Vector Biology, Liverpool School of Tropical Medicine, Pembroke Place, Liverpool L3 5QA, UK; daniel.mcdermott@lstmed.ac.uk

**Keywords:** insecticide resistance, resistance monitoring, method validation, WHO tube

## Abstract

**Simple Summary:**

The “WHO susceptibility bioassay” is a method from the World Health Organization used to monitor the resistance to insecticides in mosquito populations. This method was first developed in the 1960s and has undergone multiple changes since then. While these changes may appear minor, the numerous iterations of the test procedures leave some parameters open to interpretation, and changes to methodology may affect results. To address this, we reviewed the published test procedures for this method and the published literature which cited this method to see where the method could be optimized and exactly how people were reporting their use of this method. This revealed that the method is not being carried out consistently, and that the most up to date iterations of the test procedures are not always referenced. To address this, recommendations on the referencing and reporting of this method were developed. Alongside this literature review, we detail experimental work that explored whether altering parameters with room for interpretation in the test procedures could impact bioassay results. From the results, suggestions have been made to tighten certain parameters to avoid inaccurate measures of insecticide resistance. Closer adherence to the method and tightened parameters should lead to the generation of more robust data from the bioassay.

**Abstract:**

Accurately monitoring insecticide resistance in target mosquito populations is important for combating malaria and other vector-borne diseases, and robust methods are key. The “WHO susceptibility bioassay” has been available from the World Health Organization for 60+ years: mosquitoes of known physiological status are exposed to a discriminating concentration of insecticide. Several changes to the test procedures have been made historically, which may seem minor but could impact bioassay results. The published test procedures and literature for this method were reviewed for methodological details. Areas where there was room for interpretation in the test procedures or where the test procedures were not being followed were assessed experimentally for their impact on bioassay results: covering or uncovering of the tube end during exposure; the number of mosquitoes per test unit; and mosquito age. Many publications do not cite the most recent test procedures; methodological details are reported which contradict the test procedures referenced, or methodological details are not fully reported. As a result, the precise methodology is unclear. Experimental testing showed that using fewer than the recommended 15–30 mosquitoes per test unit significantly reduced mortality, covering the exposure tube had no significant effect, and using mosquitoes older than 2–5 days old increased mortality, particularly in the resistant strain. Recommendations are made for improved reporting of experimental parameters

## 1. Introduction

Test procedures are published by the World Health Organization (WHO) on the use of the WHO insecticide susceptibility bioassay (or WHO tube bioassay) to monitor the resistance in adult mosquitoes to a range of insecticides commonly used for mosquito control [1]. Resistance monitoring using this approach relies on the collection of wild female adult mosquitoes or wild larvae, which are then reared to adulthood in a test facility. They are then exposed to a discriminating concentration (DC) of insecticide on a treated filter paper so that their knockdown and mortality can be scored.

The WHO tube bioassay is a simple direct response-to-exposure test. The test kit was developed in 1958 to test for the emergence of resistance to organochlorine and organophosphate insecticides following widespread resistance to organochlorine insecticides [2]. It was designed to expose a defined number of adult mosquitoes of known ages and physiological statuses to an insecticide impregnated on a filter paper for a standard exposure time (1 h).

While the WHO test procedures provide parameters for some key environmental conditions which should be kept constant while carrying out the bioassay, during insecticide resistance monitoring there are multiple potential sources of (non-resistance-associated) variability, which can influence the result of the bioassay. During a recently concluded formal WHO multicenter study to establish species-specific discriminating concentrations and procedures for new and existing insecticides, WHO tube and WHO bottle assays (an adaptation of the Centre for Disease Control (CDC) bottle bioassay developed to align end points with those of the WHO tube test) were used to generate concentration response data in multiple testing centers for a range of insecticides and multiple *Anopheles* and *Aedes* species, so that DCs could be established and validated. Within this extensive dataset, a substantial degree of variation was seen both within and between centers performing replicate assays using the same standardized methodologies [3].

One of the sources of variability in data generated when using the WHO tube bioassay methodology is the mosquitoes being tested. When using the method in the field to screen for resistance in the target population of an intervention, wild-caught adult mosquitoes should be used so that any differences in susceptibility may more closely reflect the changes in intrinsic resistance level seen for a particular intervention, and the sampled populations will be representative samples of the wild vector population in terms of age distribution and genetic variability. However, the age distribution, blood feeding status, nutritional status, and gravidity will vary between samples and potentially reduce the comparability of the results between tests and between sites. This differs from the mosquito populations that DCs are established on. Moreover, while operators using this methodology should ensure that they follow the guidance for the selection of mosquitoes for testing, it is still possible to accidentally include mosquitoes that fall outside these parameters (e.g., older than 2–5 days, males, partially blood-fed) when testing with wild-caught mosquitoes. Instead of wild-caught mosquitoes, F_1_ progeny of wild-caught mosquitoes can be used; although this requires facilities to rear and test the mosquitoes, there is greater control over the rearing conditions than for wild-caught mosquitoes.

Larval rearing conditions in laboratories have been shown to have an impact on bioassay results. Overcrowding or poor diet reduce insecticide tolerance by reducing size and fitness, for example [4,5]. Any impact on longevity because of larval rearing conditions could impact the outcome on mosquito survival. Poor mosquito survival could lead to high control mortality and more discarded tests (control mortality > 20%); this will affect the feasibility of testing. Larval rearing may also be more important for insecticides such as chlorfenapyr, where the effect of the compound is impacted by metabolism, though the correlation between longevity and size is not always positive [6]. In contrast, when the WHO tube bioassay is employed for research, a well-characterized (and ideally susceptible, so that resistance levels do not have to be maintained) laboratory strain can be used alongside the mosquito strain of interest. The benefit of this is that the researcher knows the rearing conditions of their laboratory strain and the resistance status and background of the strain, while this is not the case for field-caught mosquitoes, this well-characterized susceptible reference strain can be tested alongside as a comparator.

The effect of time-of-day of testing on bioassay results is not well-explored in the literature, but *Anopheles* typically bite at night when they may be more metabolically active, and so susceptibility testing may yield differential results if conducted during the day than during the night. Most testing is carried out during the day, so this is unlikely to be a significant source of variability of the data currently being generated using the WHO tube bioassay method. It is good practice, however, to conduct resistance-monitoring assays at the same time of day each time and report the testing time alongside the data to aid interpretation, as it has been shown that time of day can impact metabolic detoxification and insecticide resistance in *Anopheles gambiae* [7]. Chlorfenapyr (while not validated for use in the WHO tube bioassay) is also strongly affected by a temperature of <25 °C [3].

Mosquito age has been shown to affect insecticide resistance, with mosquitoes older than 10 days post-emergence showing increased susceptibility to insecticides [8]. As well as age of mosquito, the nutritional status of adults can also affect the response to insecticide exposure. Machani et al. showed that the ingestion of a blood meal increased insecticide susceptibility [9]. Further to this, it has also been shown that lowering the temperature during insecticide susceptibility testing below the recommended 27 ± 2 °C can strongly affect insecticide tolerance [10]. As part of a study conducted in a Ugandan field insectary, which lacked environmental controls but where temperature and humidity were monitored, a strong and highly statistically significant decline in *A. gambiae* mortality was detected as humidity increased [11]. In light of this, it is important to be as consistent as possible when performing susceptibility bioassays and, where it is not possible to control the conditions fully, at least to understand the effect external factors can have on the outputs from this testing and report the environmental conditions alongside the data so that the results can be interpreted accordingly.

The WHO test procedures for monitoring Insecticide susceptibility have been reviewed and updated multiple times since their original publication in 1958, and some methodological details have changed between versions. We therefore set out to review the current literature to identify which test procedures are being referenced when using this method, what data and methodological detail is being reported when this method is used, and where data gaps lie for this methodology. We aimed to achieve this by looking for parameters in the test procedures which leave room for interpretation and using a literature search to explore how these parameters can influence the results of the bioassay. Parameters which were not clearly defined or supported with evidence and where evidence is not already available in the literature were chosen to be explored experimentally. In doing this, we hope to suggest additional guidance on the optimum method for performing the WHO tube bioassay, as well as the key information required for the reporting of insecticide resistance data, thus producing more robust data and reporting it in a way that supports more meaningful interpretations.

## 2. Materials and Methods

### 2.1. Test Procedures Review

Thirteen WHO documents containing details outlining how to perform the WHO tube bioassay or the rationale behind the bioassay parameters, both published test procedures and meeting reports were reviewed to extract specific methodological details outlined in the test procedures. For each of these documents, the specifics and justifications for the methodological details outlined below were noted for each document and then compared.

Mosquito ageNumber of mosquitoes per test unitNumber of mosquitoes required per treatment testedTube remaining still during exposure or being agitatedVertical or horizontal orientation of the test unitExposure timeInsecticide concentrationsCarrier oilDetails on insecticide-treated paper use, preparation (if not purchased from the WHO site in Malaysia), and storageInsecticide class specific recommendations for carrying out the bioassayAny recommendations for behavioral assessmentSpecies specific recommendations for carrying out the bioassayInterpretation of resultsDetails on inclusion of positive and/or negative controlsUse of synergistsCriteria for scoring knockdown/mortality

### 2.2. Literature Review

In January 2021, a literature search was performed on PubMed and BioMed Central databases using the search terms “mosquito”, “WHO”, and “tube” on both PubMed and BioMed Central. Results were sorted by relevance; 49 results were returned in PubMed, and 1483 results (of which a sample of the first 740 (~50%) were screened as a representative sample) were returned in BioMed Central using these search terms. Of the publications selected for inclusion in this way, 35 came from PubMed and 57 from BioMed Central. Duplicates were excluded, and further publications were excluded to allow for the comparison of methodology and mortality data between publications, including:Incorrect referencing of WHO tube bioassay test procedures (e.g., an academic publication or other WHO documentation which did not provide a fully outlined protocol for the WHO tube bioassay)Number of mosquitoes used for bioassay was not reported—the power of the study cannot be determined, so the statistical significance of the data is unknown and therefore not comparable to other studiesMosquito age was not reported—the data could be generated on mosquitoes outside the recommended testing age range and so the study is not necessarily comparable to other studies.Non-standard insecticide was used in the exposure tube for example use of technical grade chlorfenapyr, which had no DC established in the test procedures (however, a tentative DC was outlined in the 2016 test procedures [1]); use of a formulated IRS product; or a version of the bioassay adapted to use an LLIN.

A second search was performed using “mosquito”, “WHO”, “susceptibility”, and “bioassay” to account for alternative ways of referring to the bioassay, but no additional publications were identified. This left 61 publications to be included in the analysis, as detailed in Figure 1.

The same information was then extracted from the identified publications as was extracted from the test procedures as well as:Insecticide class-specific recommendations for carrying out the bioassay.Species-specific recommendations for carrying out the bioassay.Interpretations of results.If field mosquitoes were used, whether a susceptible reference strain was tested alongside.Sample size.Any additional methodological details.

Details of all identified publications are given in Appendix A.

### 2.3. Experimental Investigation of Parameters

#### 2.3.1. Mosquito Rearing

Mosquito colonies were maintained as described by Williams et al., in the LITE facility at the Liverpool School of Tropical Medicine (LSTM) [12]. Insectary conditions were maintained at 26 ± 2 °C and 70 ± 10% relative humidity (RH), with a L12:D12 h light: dark cycle and a 1 h dawn and dusk. Larvae were reared in purified water and fed ground TetraMin^®^ tropical flakes (Tetra U.S., Blacksburg, VA, USA), adults were provided continuous access to a 10% sucrose solution, and adult females were given access to blood using a Hemotek membrane feeding system (Hemotek Ltd., Blackburn, UK). Two well-characterized laboratory strains of mosquito were used as representative populations, one susceptible and one resistant to commonly used insecticides. *A. gambiae* Kisumu is a reference insecticide-susceptible strain originally from Kisumu, Kenya, reared at LSTM since 1975, and *A. gambiae* Tiassalé 13 is a resistant *Anopheles* strain which was colonized from Tiassalé, Côte d’Ivoire and has been reared at LSTM since 2013. Kisumu has no selection procedure and so is susceptible, whereas Tiassalé 13 is selected with a 1 h exposure to 0.05% deltamethrin and shows high resistance to pyrethroids, which is mediated by both target sites 1014F *kdr* and *ace-1*, and metabolic resistance, which is mediated by several cytochrome P450s.

#### 2.3.2. WHO Tube Bioassay Testing

A WHO holding-tube and its exposure tube pair are referred to as a ‘test unit’. Test units using mosquitoes from the same cohort are technical replicates of each other. Test units using mosquitoes from different cohorts are referred to as biological replicates. Three biological replicates, each made up of 2 negative control test units and 12 insecticide test units, were carried out for each experiment. There were 2 test units per treatment within a biological replicate, which were technical replicates of each other. The WHO tube bioassay 4 was used with some adaptation to allow investigation of 2 individual parameters:The number of mosquitoes per test unit—this was chosen as ~10% of publications identified in the literature review reported using less than the recommended range of mosquitoes as outlined in the test procedures, and a further ~13% did not report this information.Covered or uncovered exposure tube—this was chosen as it is mentioned only in the test procedures from 2016.Mosquito age—this was chosen as ~44% of publications identified used mosquitoes of the incorrect age for the test procedures they referenced.

Other factors, such as orientation of the test unit, sample size required per treatment, and sample size required per control, were either already clearly defined in the current test procedures or supported by previously published literature.

Grade 1 Whatman filter papers of size 15 × 12 cm were coated with 0.043% permethrin dissolved in silicone oil for Kisumu testing (an LC_50_ determined from previous work in the department by WHO tube bioassay for the Kisumu strain) and 0.75% (WHO recommended discriminating concentration for *Anopheles* [1]) permethrin for Tiassalé 13 testing. The 0.043% papers were made in the LITE laboratories, whereas the 0.75% papers were purchased from WHO Malaysia (Universiti Sains Malaysia, Penang, Malaysia). Permethrin was chosen as it is a heavily used insecticide for profiling, with a well understood mode of action and established resistance mechanisms in the Tiassalé 13 strain.

To investigate the effect of the number of mosquitoes per test unit and compare the results from covered and uncovered tubes in parallel, the experimental layout for a single replicate is outlined in Table 1. Additional technical replicates of the tubes containing fewer mosquitoes were conducted to ensure equivalent numbers of mosquitoes were screened per treatment and to nullify the potential bias of a smaller sample size influencing the mortality estimate.

Mosquitoes were exposed in test units for 1 h at 26 ± 2 °C and 70 ± 10% RH and then transferred back to holding-tubes post-exposure and held in the same conditions for 24 h at which point their mortality was scored. Data from three biological replicates, each prepared independently, were used to generate the data.

To investigate the effect of mosquito age, a second experiment was completed using a single cohort of each strain, from which a subsample of 150 mosquitoes was taken at the WHO recommended testing age [3] (2–5 days), 2 days after testing age (overlapping with the recommended age range, 4–7 days), and 4 days after testing age (outside the recommended age range, 6–9 days). Mosquitoes were tested with two negative control test units and four insecticide test units. The test concentrations were 0.043% for Kisumu testing and 0.75% for Tiassalé 13 testing, as in the previous experiment. This experiment was repeated three times, each with independently reared cohorts of mosquitoes.

#### 2.3.3. Data Analysis

Mortality was calculated as the total number of individuals knocked down or dead in a test unit as a percentage of the total number of individuals in the test unit at the end of the 24 h scoring period. If the mortality in the negative control test unit was <20% but ≥5%, then the observed mortality in each treatment test unit was corrected using Abbott’s formula [13]. Where the control mortality was ≥20%, the results were discarded and the test replicate repeated.

Both datasets (age and number of mosquitoes per tube) were screened using a binomial generalized linear model (GLM), a binomial generalized linear mixed model (GLMM) with a random effect for biological replicates to account for any inter-assay variation, a binomial GLMM with a random effect for biological replicates, and a nested random effect for technical replicate to account for the intra-assay variation using the glmmTMB package in R [14]. For each analysis, the variable was treated as a factor, with 5 days used as a reference for age and 25 mosquitoes used as a reference for the number of mosquitoes in the tube. The negative controls were excluded from all analyses, as we were only testing for the influence of these factors on the deviation from the reference. A likelihood ratio test (LRT) was conducted to identify the best-fitting GLMM.

## 3. Results

### 3.1. Comparison of Test Procedures

Although the current WHO test procedures recommend a single DC assay to detect resistance [3], the WHO tube bioassay method initially recommended a concentration response experiment [15]. Field-caught blood-fed females were used, with 15–25 mosquitoes per test unit using a series of four concentrations, which should lie on a range giving 0–100% mortality with four replicates per concentration for a total of 200 mosquitoes per test concentration. If a population of mosquitoes was highly resistant, the exposure time was increased by 1 h until significant mortality was seen. This method continued to be recommended until 1970, when the method changed from recommending four concentrations to only two concentrations, with the lowest concentration to be tested first with a range of exposure times [16]. At a WHO meeting in 1976, this was changed again to a single concentration known as the discriminating concentration [17]. These updates were made in order to simplify the bioassay to fewer test concentrations for the growing list of insecticides which required resistance monitoring.

The test kit itself was initially eight exposure tubes marked with a red dot, two control tubes marked with a green dot, and ten holding-tubes also marked with a green dot (see Figure 2). Moreover, it was specified that the impregnated papers could initially be reused up to 20 times, and the test kit had to be oriented with the mesh screen facing up during exposure.

Several meetings were held between the years 1958 and 1992 to discuss the changes to the methodology to address increased resistance to insecticides and to add new insecticide classes (See Table 2 for a list of meeting reports) [2,15,16,17,18,19,20]. The test procedures were then updated in 1998 following a multicenter study which recommended DCs for five pyrethroid insecticides [21]. This update also included some methodological changes. Single discriminating concentrations were provided for both the newly added pyrethroid insecticides and the organochlorines, organophosphates, and carbamates. Minor adjustments were made to the test kit itself to reduce it from a 20-test unit kits to a 12-test unit kit consisting of five exposure tubes marked with red dot, two control tubes marked with a green dot, and five holding-tubes also marked with a green dot. Testing a minimum of 100 mosquitoes (4–5 replicates of 20–25 mosquitoes) per concentration was recommended. Mosquitoes for testing were now required to be 1–3-day-old non-blood-fed females. These mosquitoes were either F_1_ progeny from larval collections or field-caught mosquitoes. The temperature range of 25 ± 2 °C and 70–80% RH was specified. Insecticide-treated papers were only able to be used 5 times, as opposed to the previously recommended 20. The vertical orientation of the test tubes during performance of the bioassay was further justified in these test procedures, as horizontal positioning avoids the knockdown and recovery of mosquitoes, since knocked down mosquitoes would lie on treated paper instead of the untreated mesh-end of the test unit and so still be exposed to the insecticide. This would increase the exposure of the mosquito, and the exposure route may not be through the tarsi of the mosquito [21].

The WHO test procedures were then updated again in the 2006 “Guidelines for testing mosquito adulticides for residual indoor spraying and treatment of mosquito nets”. Little changed between the 1998 version of the test procedures and this version; the recommended humidity changed from 70–80% RH to 80 ± 10%, and 2–5 day old mosquitoes were specified instead of the previous 1–3 day old [22]. Then, in 2013, the “Test procedures for insecticide resistance monitoring in malaria vector mosquitoes” was published. Minor adjustments were made to the test kit itself; the new 12-test-unit kit consisted of four exposure tubes marked with a red dot, two control tubes marked with a yellow dot, and six holding-tubes also marked with a green dot. At least 120–150 active 3–5-day old female mosquitoes were recommended to be exposed in batches of 20–25, ideally with at least 100 per insecticide and 50 as controls [23].

**Figure 2 insects-13-00544-f002:**
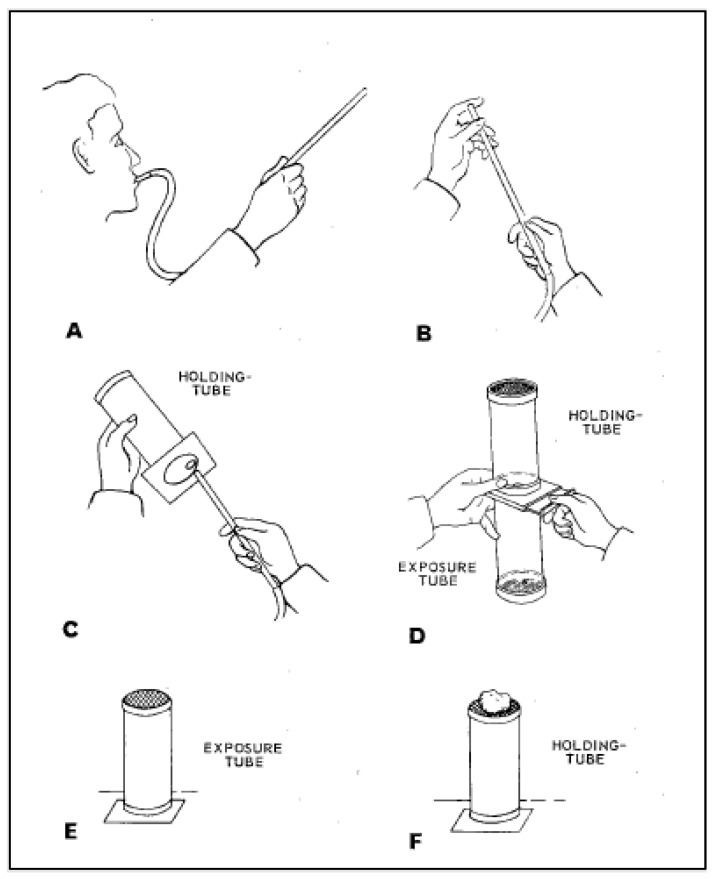
Original World Health Organization (WHO) tube method as outlined in the “8th Report of the Expert Committee on Insecticides” [24], reproduced with permission of Rajpal Singh Yadav, WHO. (**A**) Collect test mosquitoes using a mouth aspirator. (**B**) Mosquitoes should be collected in batches of no more than 10. (**C**) Test mosquitoes are gently transferred to the holding-tubes until they number 20–25 per tube. (**D**) The exposure tube is attached, and the slide is opened. Mosquitoes are then gently blown from the holding-tube to the exposure tube. The holding-tube is detached and set aside (**E**) The exposure tubes are left standing upright for 1 h during the exposure. (**F**) Mosquitoes are transferred back to the holding-tube by reversing the process described in C. The holding-tube is set upright, and a pad of wet cotton wool is placed on top. Tubes are held for 24 h, at which point mortality counts are made.

The most recent update to the test procedures came in 2016 [25]. These procedures aimed to provide a stronger focus on producing operationally meaningful data and so introduced resistance intensity (RI) assay testing (using 5× and 10× the pyrethroid DC) and pyrethroid-PBO synergist bioassays as additional testing alongside the standard WHO insecticide susceptibility bioassay. Again, slight changes were made to the WHO tube bioassay protocol. The temperature and humidity changed to 27 ± 2 °C and 75 ± 10% RH, and it was recommended that the test units be “placed in an area of reduced lighting or covered with cardboard discs”. This was supposed to reduce the light intensity and discourage mosquitoes resting on the mesh. There was also an additional piece of WHO documentation in the 2016 “Monitoring and managing insecticide resistance in *Aedes* mosquito populations Interim guidance for entomologists”, which was published as part of the response to the Zika epidemic. However, there were no methodological differences in performance of the bioassay from the previously published 2016 procedures [25]. The same methods are thus recommended for *Aedes* spp. as for *Anopheles* species.

The criteria for scoring knockdown and mortality in this bioassay have remained unchanged. However, there is room for interpretation around what is or is not a knocked down mosquito. When testing pyrethroids with adult mosquitoes, it is common to see surviving individuals with several legs missing. These mosquitoes are still technically alive and able to fly but have clearly been impacted by the exposure. To take this into account, Hougard et al. assessed “functional mortality” alongside normal mortality scoring (dead mosquitoes only). Functional mortality was defined as “including surviving mosquitoes with three legs or fewer”, as it is assumed that mosquitoes with three legs or fewer would not survive in the field. From this study, considering functional mortality provided additional information as well as a better estimate of the overall killing effect of a pyrethroid insecticide [26]. However, Isaacs et al. showed that insecticide-induced leg loss had no significant effect upon either the blood-feeding or egg-laying success of exposed mosquitoes. A non-significant reduction in blood-feeding success was seen with 1-legged insecticide-exposed mosquitoes, and, while their egg laying behavior appeared to be altered, the eggs laid were fertile and hatched to larvae. We conclude that studies of pyrethroid efficacy should not discount mosquitoes that survive insecticide exposure with fewer than six legs, as they may still be capable of biting humans, reproducing, and contributing to malaria transmission [27].

### 3.2. Review of the Literature

Only the 1998, 2006, 2013, and 2016 test procedures were referenced in the sampled publications, with the majority referencing either the 1998 or the 2013 test procedures. However, when comparing the publication date of a journal article and the publication dates of the test procedures referenced within, over half the publications were using test procedures that were between 3 and 18 years out of date (See Figure 3).

The test procedures have remained consistent since 1998 in outlining the number of mosquitoes per test unit as 20–25. When looking at the number of mosquitoes per test unit used for testing in the published literature (see Figure 4), approximately 90% were within the WHO range. Those that lay outside the range tended to use between 10 and 15 mosquitoes; these were often field studies, and so this was likely due to the limited availability of mosquitoes in the field. This was also mirrored in the number of mosquitoes used per treatment. The WHO recommend 100 per treatment, but again field studies often used less than this, which again was probably because of mosquito availability. Studies that showed numbers of mosquitoes per treatment larger than 250 were often pooled from multiple sites or multiple rounds of testing. However, 44% of the publications sampled reported “20–25” mosquitoes instead of the actual numbers used per test unit, which shows that they followed the test procedures but does not provide accurate ‘n’ values for a given treatment. Several papers reported using mosquitoes in the range of “15–25”, “10–15”, or “10” per test unit. No justification for this deviation from the WHO test procedures for this bioassay is provided within the publication. However, it can be assumed that, due to these studies either using field-caught larvae reared to adults or F_1_ larvae of field-caught adults reared to adults for their bioassay testing, they would be limited in terms of total sample size and so reduced the number per test unit to increase technical replication.

Since 2013, the WHO test procedures have recommended a minimum of 50 mosquitoes to be exposed to control papers in 2 batches of 25 each alongside the 100 required per treatment. This is often not reported in the literature, with around 80% of publications not reporting this information; however, this is unsurprising, as it is a more recent addition to the WHO test procedures.

The source of the exposure papers is often not reported, with nearly 45% of publications using terminology along the lines of “papers impregnated with insecticide were used”; however, it is unclear from this whether papers were made by the researchers themselves or purchased from Malaysia. Since 1993, the WHO have provided standardized insecticide papers from their site in Malaysia, and over 40% of publications stated that their exposure papers were sourced from there. The studies which did specify the source of papers as other than from the WHO either impregnated their own exposure papers, had them made up by a partner research institute, or purchased them from a center for disease control or other public health body.

The recommended mosquito age has changed several times throughout the different iterations of the test procedures. In 1998, 1–3 days was recommended, until this was updated in 2006 to 2–5 days and again in 2013 to 3–5 days. For publications referencing the 1998 test procedures, 85% used mosquitoes older than recommended. For publications referencing the 2013 test procedures onwards, 23% used mosquitoes younger than recommended. So, 44% were using mosquitoes of the incorrect age for the test procedures they referenced (see Figure 5).

The sampled manuscripts described the results from a range of Anopheline and Culicine species, though the species were not always identified, as well as a large number of insecticides from different mode of action classes (detailed in Appendix A). In instances where more than one publication tested the same combination of strain of mosquito and insecticide, we compared the data between the two publications. A total of 44% of publications used only a field strain and so data was not comparable. For the remaining publications, 38% included a susceptible *A. gambiae* (Kisumu), 13% used an unspecified laboratory strain, two publications used a susceptible *A. funestus* (FANG), and one publication used the susceptible *A. coluzzii* (N’gousso) as reference strain alongside the testing of field populations. The data for these susceptible reference strains agreed between the publications; however, the mortality was often 100%, as the strain being tested was a susceptible laboratory strain. Three publications exposed resistant mosquito strains to discriminating concentrations to profile their resistance phenotype. Bagi et al. [28] and Williams et al. [12] both exposed Tiassalé 13 to 0.75% permethrin for 1 h; Bagi showed a 3.4% mortality 24 h post-exposure, whereas Williams et al. showed ~20% mortality when the strain was profiled in the years 2017 and 2019 [12,28]. Owusu et al. [29] also exposed Tiassalé 13 to 0.75% permethrin for 1 h and showed a mortality of 78.0%, whereas Williams et al. [12] showed approximately 5% mortality for the years 2017 and 2019.

### 3.3. Experimental Investigation of Parameters

The GLMM accounting for biological effect was used to generate the effect estimate for the two variables of interest. The Kisumu strain was much more susceptible to knockdown, assessed at 60 min, when there was a reduction in the number of mosquitoes per tube with a significant reduction in knockdown for tubes with 20 (OR = 0.42, *p* = 0.001, 95% CI = 0.26–0.69), 15 (OR = 0.35, *p* ≤ 0.001, 95% CI = 0.21–0.59), and 10 (OR = 0.2, *p* ≤ 0.001, 95% CI = 0.11–0.36) mosquitoes. This significant reduction was still found when evaluated again at 24hrs in the tubes of 10 and 20 mosquitoes; however, this was no longer present for the tubes containing 15 mosquitoes. For the Tiassalé 13 data, the 60 min assessment also found a significant reduction in tubes containing 10 mosquitoes (OR = 0.28, *p* = 0.004, 95% CI = 0.12–0.67); however, this effect was not discernable when evaluated again at 24 h (Figure 6, Appendix B).

An additional treatment was performed with 25 mosquitoes per test unit with a cardboard disc covering the top of the tube during exposure. No significant difference was detected for this alteration in the study protocol in either the Kisumu strain (OR = 0.83, *p* = 0.391, 95% CI = 0.53–1.28) or Tiassalé 13 at 24 h (OR = 1.27, *p* = 0.462, 95% CI = 0.67–2.39).

For the Kisumu strain, the 4–7-day-old and 6–9-day-old mosquitoes showed a significant increase in mortality at 60 min knockdown. However, only the 6–9-day-old mosquitoes maintained this statistical significance when assessed at 24 h (OR = 2.46, *p* ≤ 0.001, 95% CI = 1.69–3.59). The mortality assessment for the 2–5-day group’s mortality increased from around 26 to 64% between assessment periods.

This trend for older mosquitoes to show a greater susceptibility following exposure was also seen for the Tiassalé 13 strain (Figure 7) with the 6–9-day-old group showing an increase in mortality of ~9% points compared to the 2–5-day-old group at 60 min (OR = 2.99, *p* ≤ 0.001, 95% CI = 1.64–5.46), and both the 4–7-day- and 6–9-day-old groups showing increased mortality at 24 h (Figure 7).

## 4. Discussion

It is clear from the details outlined in Table 2 that the WHO susceptibility bioassay has undergone numerous updates to its methodology since its inception. While at each stage these updates have been relatively minor, it is still possible that these could impact bioassay results, and so it is important to ensure that the most recent iteration of the test procedures is followed and referenced. However, the literature review shows that this is not always the case.

The methodological variability between the published test procedures and the way these test procedures were historically presented on the WHO website leaves the WHO tube assay for insecticide susceptibility in mosquitoes open to interpretation as to how to perform the bioassay, as well as being unclear as to what the most up to date iterations of the guidelines are. The WHO website has been updated since this review of the method began, and the relevant test procedures can now be found considerably easier (https://www.who.int/teams/global-malaria-programme/prevention/vector-control/insecticide-resistance [Accessed: 4 February 2022]). Moreover, the lack of comparable data from the published literature is due to the populations being tested being either field strains of unknown resistance status or a susceptible laboratory strain. For the few publications where the same resistant laboratory strains were able to be compared, the mortality data was wildly different. This could be because the same strains held in different labs might in fact be vastly different from each other. This could be because of a whole host of reasons, including laboratory adaptations, contamination, selection pressure applied rearing conditions, genetic bottlenecks, and genetic drift. As a result, to optimize this bioassay, we planned to investigate the effect of mosquito age in the range of 5–10 days old, as well as the number of mosquitoes per test unit and the use of cardboard discs to cover the mesh of the exposure tube during the test, which is specified in the most recent iteration of the test procedures. These factors were chosen for investigation, as there is a lack of published literature investigating their effect on the outcome of this bioassay.

The susceptible *A. gambiae* strain Kisumu was exposed to permethrin-treated papers of a concentration expected to provide moderate mortality in an experiment to explore the effect of varying the parameters of interest when conducting the WHO tube assay. Based on three replicate tests, there was no evidence that covering the top of the exposure tubes with a cardboard disc during the exposure period had any impact on either 1 h knockdown or 24 h mortality. The rationale for the covering of the exposure tubes using in the test procedures is that it will prevent light entering through the mesh and so should discourage mosquitoes from resting on the upper mesh of the test units during exposure, which reduces their contact with the insecticide. It was not possible to assess if there was a reduction in resting on the mesh, as it was not possible to observe mosquito behavior during the exposure period, as the exposure chamber was covered by the insecticide-treated filter paper and the cardboard disc. However, due to the lack of significant difference in mortality seen in this study, we would suggest that this step appears to be unnecessary. So long as all test units are treated the same in terms of lighting, mosquitoes resting on the mesh should be consistent between test units and therefore there should have no impact on the final mortality scoring.

When varying the number of mosquitoes per test unit, mortality in this same experimental set up was unaffected by mosquito numbers between 15 and 30 mosquitoes per test unit. However, when only 10 mosquitoes were added per test unit, the 24 h mortality was significantly lower (7% compared to 50%). The same trend is not seen in the proportion of mosquitoes knocked down immediately post exposure, with knockdown being reduced in treatments with 15 and 20 mosquitoes per test unit compared to covered and uncovered treatments containing 25 or 30 mosquitoes. Knockdown thus appears to be positively correlated with the number of mosquitoes per test unit in this laboratory strain. This implies that mosquitoes are being differentially exposed during the bioassay, depending on the number of individuals within a single test unit. It is possible that, when using 10 mosquitoes in a test unit, there is enough space for all or most of the mosquitoes to rest on the door at the base or at the mesh at the top of the test unit and therefore avoid contact with the insecticide-treated paper. With more mosquitoes, there is more opportunity for this free flight to disturb resting mosquitoes within the bioassay and cause them to fly and resettle in a different part of the exposure chamber. This could then force the mosquitoes which were previously resting on a non-insecticide-treated surface to encounter the insecticide-treated filter paper instead and become intoxicated with the insecticide. The more mosquitoes in a test unit, the more likely this disturbance is to occur and, in turn, the more likely a mosquito is to become intoxicated with insecticide through more frequent contacts. We recommend that at least 15 mosquitoes are included per test unit when conducting WHO tube assays, and that, where knockdown is the entomological endpoint of interest, the number of mosquitoes per test unit is held constant between replicates and treatments. As well as this, due to the general increase in mortality seen with the addition of more mosquitoes, we would not recommend exceeding 30 mosquitoes per test unit. It is also worth noting that there was a substantial divergence in the mortality estimate for biological replicate test compared to the other three replicates. Despite the quality control measures in place, this may be due to some difference in the cohort of mosquitoes being used. It does highlight the variability that can be introduced into the bioassay data by minor changes in parameters, even in highly controlled conditions. Such a divergent bioassay result could easily go unnoticed in the absence of technical and biological replicates.

Since this data was generated from three biological replicates with the same number of technical replicates per treatment, the sample size for each treatment differs, with three times as many mosquitoes tested in the 30-mosquitoes-per-test-unit treatment as in the treatment with 10 mosquitoes. To account for this difference, an additional biological replicate was carried out to equalize the sample size for each treatment to ~180. With this additional replication, the trends seen did not change. The only significant change was that the 24 h mortality for the 10-per-test-unit treatment increased from 7 to 15%. The variability of the results within each treatment was either unaffected or reduced with increased replication, showing the value of maximizing both test unit replicates and mosquito ‘n’ values when generating data using the WHO tube assay. For the Tiassalé 13 strain in the 15-per-test-unit treatment, the variability was significantly higher (Figure 7); however, this is due to the large intraspecific variation between the technical replicates in one of the biological replicates. It is possible that using more test units with fewer mosquitoes could result in a lower variability than fewer test units with larger numbers, and so, in settings where mosquitoes are less available, it may be beneficial to divide the cohort up into multiple smaller batches with more test units. Where possible, we would recommend increased replication with different cohorts of the same mosquito population to increase the sample size to at least the WHO recommended 100 mosquitoes per insecticide treatment and 50 mosquitoes per control. We also would not recommend using any more than 30 mosquitoes per test unit, as there is no data available, that we are aware of, to support it, and we can see no logistical reason for using more than this number per test unit.

Repeating the experiment with the insecticide-resistant Tiassalé 13 exposed to permethrin-treated DC papers with results based on three replicates showed similar trends to those using Kisumu. Covering the exposure tube during the period of exposure again had no significant effect on knockdown or mortality relative to the uncovered test units. Mosquito numbers between 15 and 30 mosquitoes per test unit did not affect knockdown or mortality, but again the treatment with only 10 mosquitoes per test unit resulted in lower mortality. The effect size of the number of mosquitoes per test unit seemed to be smaller than in the susceptible Kisumu strain, and a larger sample size was needed to be able to detect a difference. Further replicates of the experiment to ensure equal ‘*n*’ values for the number of mosquitoes tested per treatment did not affect the trends of results; the variability was reduced, though not to the same extent as it was in the Kisumu experiment. This replicated result, even with a DC assay with insecticide-resistant mosquitoes, which are more field relevant than an old laboratory colony, supports the recommendation to use a minimum of 15 mosquitoes per test unit but that covering the exposure tube does not have an effect.

When investigating the effect of mosquito age at the time of testing, we found that mosquitoes both 2 and 4 days older than the recommended testing age (2–5 days) show an increased susceptibility to permethrin. This increased susceptibility is seen at 6–9 days old for Kisumu and 4–7 days old for Tiassalé 13. This difference could be due to the increased fitness cost caused by resistance mechanisms in the Tiassalé 13 strain compared with the susceptible Kisumu strain. This supports previous findings that mosquitoes aged 10 days and above show an increased susceptibility to insecticides [8,9], but there has previously been little data on mosquitoes aged between 5 and 10 days post-eclosion. While investigations of the effect of insecticides on malaria transmission focus on older female *Anopheles* which are the vectors of malaria, it is useful for other testing purposes to know across what age range mosquitoes can be used for testing and still produce the same result. When monitoring a population for the emergence of resistance, it is important that variables including ag, and mosquito density, as discussed above, are held constant to allow robust comparisons between test replicates and to allow true changes in test results over time to be identified. The data from this study suggest that, when performing a WHO tube assay, the recommended testing age of 2–5 days should be adhered to.

Intertest variability (between biological replicates) was generally larger than or similar to intra-test variation (between test units within a biological replicate) for both strains. It is possible that using more test units with fewer mosquitoes could result in a lower variability than fewer test units with larger numbers, and so, in settings where mosquitoes are less available, it may be beneficial to divide the cohort up into multiple smaller batches with more test units, within the limits of 15–30 mosquitoes discussed above.

## 5. Conclusions

While this study uses only one insecticide and one mosquito species, the insecticide chosen is a heavily used insecticide for profiling with a well-understood mechanism of action and effect on mosquito populations. The strains tested are also well-established and well-characterized laboratory strains, one of which is wholly susceptible and another of which is highly resistant. With variable results seen for this combination of insecticide and mosquito, it is possible that even more variable results would be seen with a more moderately resistant strain and with novel or less well-understood mode of action with changes to the investigated parameters.

As a result, we make the following recommendations for this bioassay method:Better reporting of the conditions that a bioassay is carried out under, including information on:Holding/exposure temperatureHolding/exposure humiditySource of insecticide-treated papersExpiry date or batch numberReporting of negative control dataReporting of total N per treatmentReporting of number of mosquitoes per test unit.All bioassay testing should be carried out with WHO tubes positioned vertically, as stated in the test procedures, to avoid increased contact with the insecticide-treated surface from knocked down mosquitoes in a horizontally oriented test unit.A minimum of 15 and a maximum of 30 mosquitoes should be tested per test unit.Use of a characterized reference strain alongside bioassay testing of field strains is recommended where possible.Cardboard discs to cover exposure tubes do not appear to be required, and this step could be removed from the test procedures. However, for consistency of results and methodological practice, it would be good to continue this until more comprehensive data is generated on the effect of light intensity on bioassay outcomes.Historical updates and discussions of the test procedures should be clearly marked as such and should ideally link to the most recent version of the test procedures to prevent poor referencing for this methodology.

## Figures and Tables

**Figure 1 insects-13-00544-f001:**
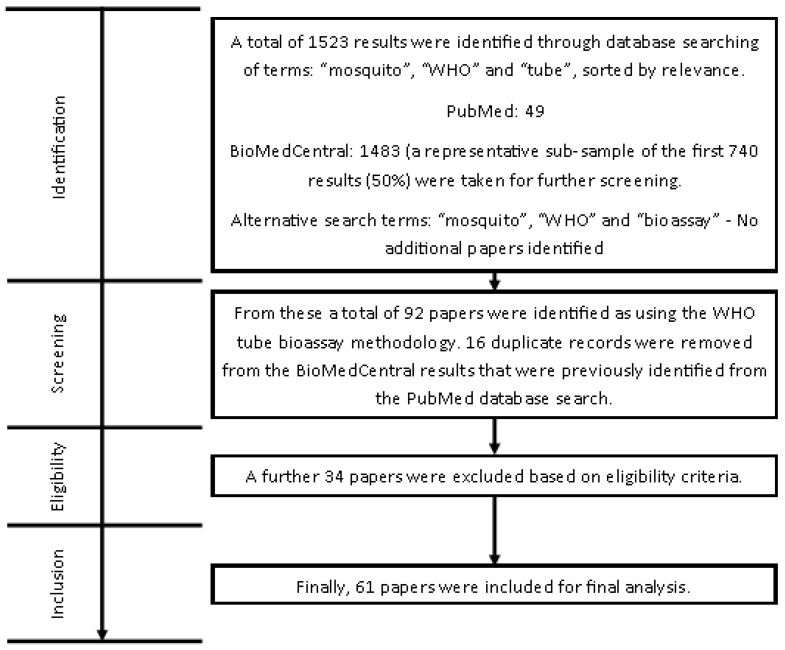
Flow diagram describing methodology of the literature review.

**Figure 3 insects-13-00544-f003:**
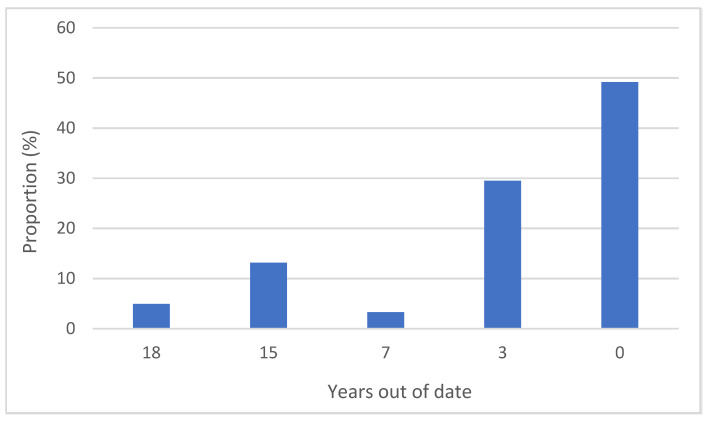
The number of years out of date the referenced guidelines were for a given publication in relation to the most recent guidelines available at the time of publication.

**Figure 4 insects-13-00544-f004:**
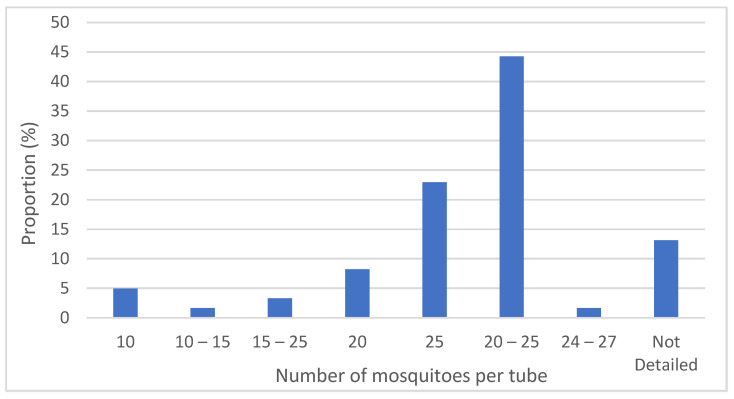
The number of mosquitoes used in an individual tube for publications reviewed.

**Figure 5 insects-13-00544-f005:**
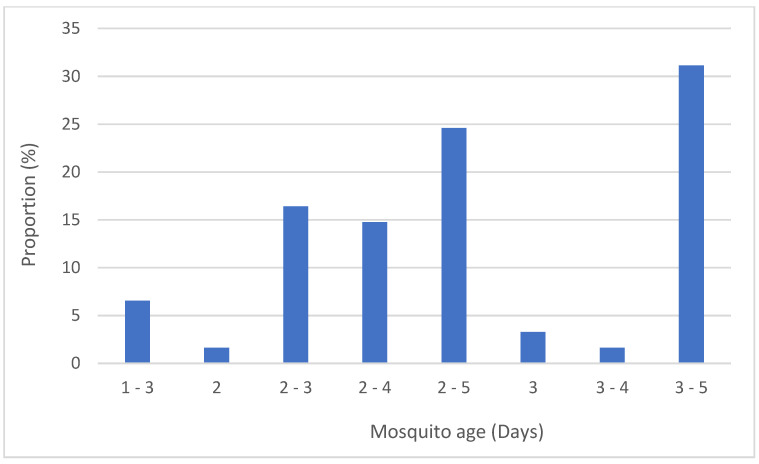
The age of mosquitoes tested for publications reviewed.

**Figure 6 insects-13-00544-f006:**
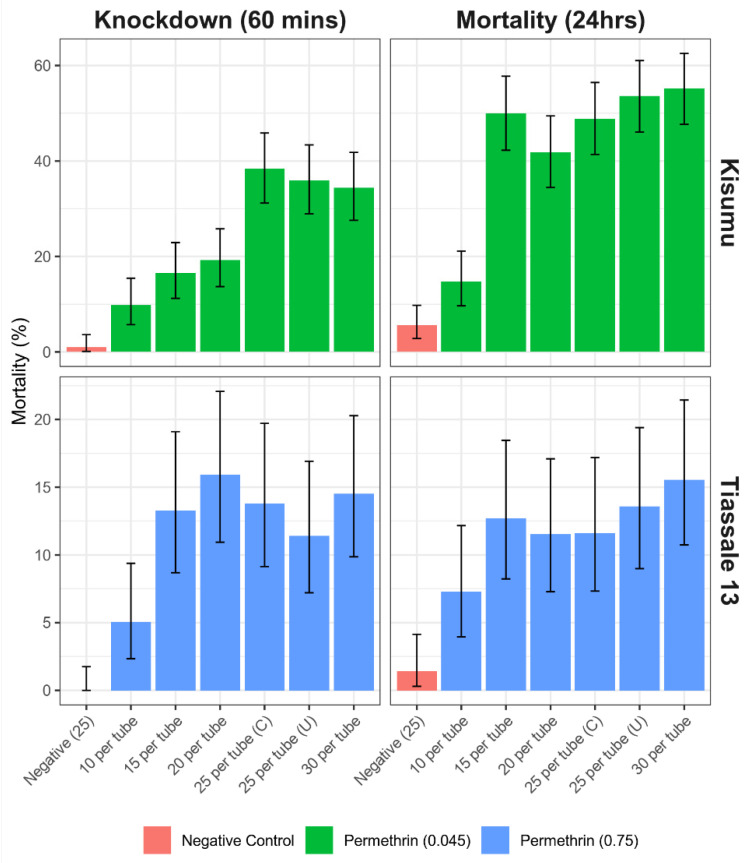
Bioassay data looking at the number of mosquitoes used in testing may impact the result of standardized bioassay testing for an *Anopheles gambiae* susceptible Kisumu strain and a resistant Tiassalé 13 strain. A total of 25 per tube (C) had the top of the tube covered during exposure, while 25 per tube (U) had the same conditions as the other tubes. Error bars equate to the 95% confidence intervals of the proportion.

**Figure 7 insects-13-00544-f007:**
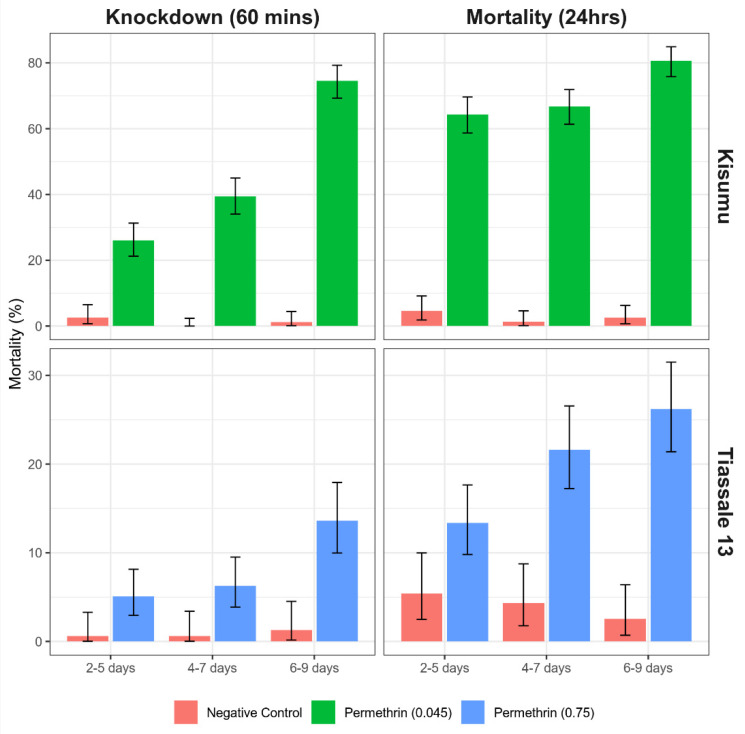
Bioassay data looking at how age of mosquitoes used in testing may impact the result of standardized bioassay testing for a susceptible *Anopheles gambiae* Kisumu strain and a resistant Tiassalé 13 strain. Error bars equate to the 95% confidence intervals of the proportion.

**Table 1 insects-13-00544-t001:** Experimental outline of a single biological replicate to investigate the effect of covered vs. uncovered exposure tubes and number of mosquitoes per test unit. The test concentration was 0.043% for *Anopheles gambiae* (Kisumu, susceptible) testing and 0.75% for *An. gambiae* (Tiassalé 13, resistant) testing. Mosquito age was 2–5 days for both strains and all treatments.

Test Unit	Treatment	Number per Test Unit	Covered/Uncovered
Negative Control 1	Silicone oil only	25	Uncovered
Negative Control 2	Silicone oil only	25	Uncovered
Uncovered 1	Permethrin	25	Uncovered
Uncovered 2	Permethrin	25	Uncovered
Covered 1	Permethrin	25	Covered
Covered 2	Permethrin	25	Covered
30 per test unit 1	Permethrin	30	Uncovered
30 per test unit 2	Permethrin	30	Uncovered
20 per test unit 1	Permethrin	20	Uncovered
20 per test unit 2	Permethrin	20	Uncovered
15 per test unit 1	Permethrin	15	Uncovered
15 per test unit 2	Permethrin	15	Uncovered
10 per test unit 1	Permethrin	10	Uncovered
10 per test unit 2	Permethrin	10	Uncovered

**Table 2 insects-13-00544-t002:** Summary of the review of historic versions of the World Health Organization (WHO) tube bioassay guidelines. * Initial baseline dose response for population generated with range of concentrations. Subsequent testing uses four concentrations along this range and two controls. Testing is performed in duplicate. ** Mortality in a negative control over 20% is unsatisfactory; testing should be repeated. A section to help with interpreting dose response curves is also introduced. *** Initial baseline preliminary test performed with full range, then concentrations selected for baseline assessment. Subsequent routine checks used a concentration which is double the concentration that has consistently given complete kill in successive tests. Exposure time can be increased for exceptionally insensitive populations. (2, 4 or 8 h) **** Only concentrations provided. Lowest concentration tested first with range of exposure times (30 min, 1, 2 and 4 h). ***** Introduction of a discriminating dose. ****** Piperonyl butoxide (PBO) synergism bioassay method included. Table is reproduced in Appendix A.

Title and Year	Mosquitoes Per Tube	Mosquito Age	Mosquito Physiological Status	Temperature	Humidity	Orientation	Insecticides	Number per Control	Number per Treatment	Lighting	Paper Usage
7th Report of the Expert Committee on insecticides (1957)	No methodological details contained within this report. Refers to the methods of “Busvine and Nash” and “Fay et al.” as possible test methods to be used for detecting and measuring resistance. No references given for these papers.										
5th Report of the Expert Committee on insecticides (1958) *	20–25	Not specified	Recently fed. Or mix of unfed and fed	Does not exceed 30 °C	Not Specified	Vertical	DDT and Dieldrin	Not specified	200	Moderate diffuse illumination.	Up to 20 times and up to weeks after opening of package Store in cool place do not refrigerate.
10th Report of the Expert Committee on insecticides (1960) **	15–25	Not specified	Recently fed. Or mix of unfed and fed	Does not exceed 30 °C	Not Specified	Vertical	DDT and Dieldrin	Not specified	200	Moderate diffuse illumination.	Up to 20 times and up to weeks after opening of package Store in cool place do not refrigerate. Expiry on package presupposes that the packages are kept sealed.
13th Report of the Expert Committee on insecticides (1963) ***	at least 15	Not specified	Recently fed. Or a mix of unfed and fed	Does not exceed 30 °C	Not Specified	Vertical	DDT, dieldrin, malathion, fenthion	Not specified	200	Moderate diffuse illumination.	Up to 20 times and up to weeks after opening of package. Store in cool place do not refrigerate. Expiry on package presupposes that the packages are kept sealed.
17th Report of the Expert Committee on insecticides (1970) ****	15–25	Not specified	Recently fed. Or mix of unfed and fed	Does not exceed 30 °C	Not Specified	Vertical	DDT, dieldrin, malathion, fenthion, OMS-33	Not specified	200	Holding tubes should be kept post exposure in a secluded shaded place.	Up to 20 times and up to weeks after opening of package. Store in cool place do not refrigerate. Expiry on package presupposes that the packages are kept sealed.
22nd Report of the Expert Committee on insecticides (1976) *****	15–25	Not specified	Recently fed. Or mix of unfed and fed	Does not exceed 30 °C	Not Specified	Vertical	DDT, dieldrin, malathion, fenitrothion, fenthion, propoxur	Not specified	200	Holding tubes should be kept post exposure in a secluded shaded place.	Up to 20 times and up to weeks after opening of package Store in a cool place do not refrigerate. Expiry on package presupposes that the packages are kept sealed.
5th Report of the WHO Expert Committee on Vector Biology and Control: Resistance of vectors of disease to pesticides (1980)	Not specified	Not specified	Not specified	~25 °C	Not Specified	Vertical	DDT, dieldrin, malathion, fenitrothion, chiorphoxim, permethrin, decamethrin, propoxur	Not specified	Not specified	Not specified.	Not specified.
10th Report of the WHO Expert Committee on Vector Biology and Control: Resistance of vectors of disease to pesticides (1986)	Not specified	Not specified	Not specified	Not specified	Not Specified	Vertical	chiorphoxim, DDT, deltamethrin, dieldrin, fenitrothion, malathion, permethrin, propoxur	Not specified	Not specified	Not specified.	Can be refrigerated so long as boxes are fully sealed.
15th Report of the WHO Expert Committee on Vector Biology and Control: Vector Resistance to Pesticides (1992)	Not specified	Not specified	Not specified	Not specified	Not Specified	Vertical	DOT, dieldrin, fenitrothion, fenthion, malathion, propoxur, lambda-cyhalothrin, permethrin, deltamethrin	Not specified	Not specified	Not specified	Not specified.
Report of the WHO Informal Consultation Test Procedures for insecticides Resistance Monitoring in Malaria Vectors, sio-Efficacy and Persistence of insecticides on Treated Surfaces (1998)	20–25	1–3 days	Unfed females	25 ± 2 °C	70–80%	Vertical	permethrin, deitamethrin, lambda- cyhalothrin, cyfluthrin, etofenprox, DDT, dieldrin, malathion, fenitrotion, propoxur, bendiocarb	Not Specified	min 100, 4–5 replicates of 20–25	Not specified	Up to 20 times.
Guidelines for Testing Mosquito Adulticides for indoor Residual Spraying and Treatment of Mosquito Nets (2006)	20–25	2–5 days	Unfed females	25 ± 2 °C	70–80%	Vertical	Not Specified	Not Specified	min 100, 4–5 replicates of 20–26	Not specified	Not more than 5 times.
Test Procedures for insecticide resistance monitoring in malaria vector mosquitoes (2013)	20–25	3–5 days	Unfed females	25 ± 2 °C	80% ± 10%	Vertical	Dieldrin, DDT, malathion, fenitrothion, propoxur, bendiocarb, permethrin, deltamethria, lambdacyhalothrin, cyfluthrin, etofenprox	50 per control, 2 replicates of 25	120–150, 6 reps 20–25 of which 2 are negative controls	Not specified.	Not more than 5 times.
Test Procedures for insecticide resistance monitoring in malaria vector mosquitoes: Second edition (2016) ******	20–25	3–5 days	Unfed females	27 ± 2 °C	75% ± 10%	Vertical	Bendiocarb, carbosulfan, propoxur, DDT, dieldrin, fenitrothion, malathion, pirimiphos- methyl, alpha-cypermethria, cyfluthrin, deltamethrin, etofenprox, lambda-cyhalothrin, permethrin, PBO	50 per control, 2 replicates of 25	120–150, 6 reps 20–25 of which 2 are negative controls	Tubes must be of places in anarea of reduced lighting or covered with cardboard discs.	Not more than 5 times.

## Data Availability

Data are contained within the article or Appendix A.

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
