# Peer review of "Reviewing the WHO Tube Bioassay Methodology: Accurate Method Reporting and Numbers of Mosquitoes Are Key to Producing Robust Results"

_insects, 2022, doi:10.3390/insects13060544_

Round 1

Reviewer 1 Report

The manuscript is about Reviewing the WHO tube bioassay methodology: Accurate  methods reporting and number of mosquitoes key to producing robust results. It is an interesting study that would be useful for several laboratories that are currently monitoring the insecticide resistance of molecules that are still available and that are heavily used to fight against insect borne disease vectors such as malaria and dengue vectors. The study included therefore a review of bioassay methodology and showed results obtained from some experimental work. Although this is relevant for most of medical entomologists, we found the statistically analysis performed very weak. To understand what was the main factor that impact bioassay results when several factors were assessed together of in parallel, we recommend authors to use mixed models with appropriate distribution to analyse their  data sets.

Specific comments:

Abstract:

Line36……… “significantly reduced mortality…..” without statistics…see Data Analysis and Results sections.

  1. Introduction:

- We found some of the sentences too long. Please consider shortening them throughout the ms. For example, see sentence at the lines 68-73.

  1. Materials and Methods

Line 198. Add a title right before ‘Mosquito rearing’.. we suggest “Experimental investigation of parameters”

Lines205-208: Add since when the colony was maintained in the laboratory prior to the set of experiments were performed.

Line 238: be consistent in the way to write numbers (0-11) and beyond in letters or numbers. Similarly, ‘hours’ or ‘hr’? This is inconsistent throughout the ms.

Line 250. Add ‘for 2-5-day-old mosquitoes’ at the end of the title. Then, remove the ‘Mosquito Age’ column in the table

Data Analysis:

Please use mixed models or consult a statistician to check the effect of biological and technical repetitions effect as well as the effect of density, age,… on the quality/robustness of bioassay results.

Line 259. Check the reference 13. Brackets are missing. Add dot after the reference.

  1. Results

Line 287. Figure2. The title does not stand alone. Please explain the meaning of letters A, B, ….F accordingly.

Lines 338, 344. Check the way the references were inserted in the text and change unless they follow the Insects guidelines. In addition, it seems like Guillet and Donnelly are the last authors of the references cited… Please check

Line 373. Check the references mentioned here and make sure they are correctly listed in the references list. Same remark for lines 414-418.

Lines 408-410….’Anopheles’ should be ‘An.’ instead for consistency

Line 404. Move ‘days’ from the numbers and put it in the X-axis title ‘Mosquito age (days)’

Figures 6 and 7. When repeating ‘per test unit’ everywhere? Please do it once in the title.

Line445. Write ‘Anopheles’ in full in the titles

Lines 443 and 464. Remove ’s’ after ‘hrs’. Same remark for Figure 8.

Line 478: “This is due to large intraspecific variation between the technical repli-478 cates in one of the biological replicates.” This sound like a discussion…Please move it where appropriate.

Line 482. Species names should be italicized

4.Discusssion.

Line 507-510. For “lab’, we suggest ‘laboratory’ instead throughout the ms.

Line 518. We suggest that ’Anopheles’ be ‘An.’ instead

Line521. ‘three’ … consistency needed throughout to use letters or numbers

Line529. Put ‘,’ after ‘However’

Author Response

We thank the reviewer for their kind words, and for engaging with the manuscript to make some very useful comments. We have addressed all comments (please see below), and believe that the additional statistical analysis suggested has strengthened the manuscript.

Abstract:

Line36……… “significantly reduced mortality…..” without statistics…see Data Analysis and Results sections.

– Further statistical analysis has confirmed the significance.

Introduction:

- We found some of the sentences too long. Please consider shortening them throughout the ms. For example, see sentence at the lines 68-73.

– ms reviewed for sentence length

Materials and Methods

Line 198. Add a title right before ‘Mosquito rearing’.. we suggest “Experimental investigation of parameters”

– Title added.

Lines205-208: Add since when the colony was maintained in the laboratory prior to the set of experiments were performed.

– additional colony details added

Line 238: be consistent in the way to write numbers (0-11) and beyond in letters or numbers. Similarly, ‘hours’ or ‘hr’? This is inconsistent throughout the ms.

– adjusted to be consistent throughout ms.

Line 250. Add ‘for 2-5-day-old mosquitoes’ at the end of the title. Then, remove the ‘Mosquito Age’ column in the table

– table adjusted

Data Analysis:

Please use mixed models or consult a statistician to check the effect of biological and technical repetitions effect as well as the effect of density, age,… on the quality/robustness of bioassay results.

– Both datasets (age and number of mosquitoes per tube) were screened using a binomial generalised linear model (GLM), a binomial generalised linear mixed model (GLMM) with a random effect for biological replicates to account for any inter-assay variation and a binomial GLMM with a random effect for biological replicates and a nested random effect for technical replicate to account for the intra-assay variation using the glmmTMB package in R. For each analysis the variable was treated as a factor with 5 days used as a reference for age and 25 mosquitoes used as a reference for the number of mosquitoes in the tube. Negative controls were excluded from all analyses as we were only testing for the influence of these factors on the deviation from the reference. A likelihood ratio test (LRT) was conducted to identify the best fitting GLMM.

Line 259. Check the reference 13. Brackets are missing. Add dot after the reference.

– Abbot’s reference added with Zotero

Results

Line 287. Figure2. The title does not stand alone. Please explain the meaning of letters A, B, ….F accordingly.

– Figure legend updated to explain individual panels.

Lines 338, 344. Check the way the references were inserted in the text and change unless they follow the Insects guidelines. In addition, it seems like Guillet and Donnelly are the last authors of the references cited… Please check

– references corrected to first named authors

Line 373. Check the references mentioned here and make sure they are correctly listed in the references list. Same remark for lines 414-418.

– Anopheles is written out in full the first time it is mentioned in each figure and table legend.

Lines 408-410….’Anopheles’ should be ‘An.’ instead for consistency

– Updated throughout ms

Line 404. Move ‘days’ from the numbers and put it in the X-axis title ‘Mosquito age (days)’

– Figure updated

Figures 6 and 7. When repeating ‘per test unit’ everywhere? Please do it once in the title.

– Figures updated

Line445. Write ‘Anopheles’ in full in the titles

– Anopheles is written out in full the first time it is mentioned in each figure and table legend.

Lines 443 and 464. Remove ’s’ after ‘hrs’. Same remark for Figure 8.

– Adjusted throughout ms

Line 478: “This is due to large intraspecific variation between the technical repli-478 cates in 1 of the biological replicates.” This sound like a discussion…Please move it where appropriate.

– Moved to discussion

Line 482. Species names should be italicized

– Figure legends formatted in line with journal style

4.Discusssion.

Line 507-510. For “lab’, we suggest ‘laboratory’ instead throughout the ms.

– Updated throughout ms

Line 518. We suggest that ’Anopheles’ be ‘An.’ Instead

  • Updated throughout ms – spelt out in full first time it is used in the main text and in each legend, and abbreviated thereafter

Line521. ‘three’ … consistency needed throughout to use letters or numbers

– Numbers one to ten updated to numbers throughout unless at the start of a sentence.

Line529. Put ‘,’ after ‘However’

– Comma added

Reviewer 2 Report

This study reviewed published test procedures for the WHO tube bioassay used in mosquito insecticide resistance monitoring to assess how this method could be optimized and how researchers were reporting its use. In addition, authors tested whether parameters that are not consistently replicated in the test procedures, such as covering or uncovering of the tube end during exposure, number of mosquitoes tested and mosquito age, could impact bioassay results. The review of published test procedures and literature showed that the method is not conducted consistently, and that most up to date procedures are not always referenced. Furthermore, experimental testing showed that two parameters, number of mosquitoes per test unit and mosquito age, affected mortality, and, therefore, the accuracy of the bioassay.  Authors made recommendations to prevent inaccurate measures of insecticide resistance and generation of robust data.

General comments:

There is a great need to standardize methodologies used in mosquito insecticide resistance monitoring, and this study focuses on a widely used insecticide susceptibility bioassay, the WHO tube bioassay, and proposes to identify sources of methodological variability through a literature review and experimental testing of parameters not clearly defined.  Overall, the approach is solid, and results support conclusions, however, selection of species and insecticide for experimental testing should be discussed in the materials and methods section rather than in the conclusion section.  Also, in the literature review section, it might be helpful to mention which mosquito species and insecticides were evaluated in the 61 publications included for analysis. Regarding bioassay data presentation in figures, it is a bit confusing to include the covered and uncovered treatments, given that is it a parameter that could be presented separately from other two parameters evaluated (mosquito age and number).  Given that mosquito age and number are reported as key parameters to obtain consistent results, would it be possible to perform statistical analyses that show that differences in mortality recorded are actually significant?  Some minor inconsistencies in unit format used (Ex: hr. or hours?), spaces between temperature and RH values, and scientific names not italicized were noticed.

Specific comments:

Introduction:

Line 54: either “hr.” or “hour” should be used consistently across the text.

Line 110: “et al.”

Materials and Methods:

Line 197: Appendix A not found in supplementary files

Line 208: gene names (kdr, ace-1) should be italicized

Lines 227-228: Briefly listing other factors could be helpful.

Lines 238-240: either “hr.” or “hour” should be used consistently across the text. Spaces between temperature and RH% values should be consistent across the text.

Table 1: check for missing periods and spaces between words.

Results:

Figure 5:  the word “days” could be deleted from the X-axis values and include in parenthesis in the X-axis label to reduce clutter.

Figures 6 and 7:  may consider removing the covered (25 per test unit) treatment so that only uncovered treatments are compared. The covered vs. uncovered comparison could be just mentioned in the text, or presented in a separate graph.  Scientific names in figure legends have to be italicized.

Figure 8: Scientific names in figure legends have to be italicized.

Author Response

We thank the reviewer for engaging with the manuscript and providing such useful comments, which have all been addressed in the revised manuscript. We think that the inclusion of additional statistical analysis in particular has helped to strengthen the manuscript.

…selection of species and insecticide for experimental testing should be discussed in the materials and methods section rather than in the conclusion section. 

  • The reasons for selecting the strains and chemistry have been added to the Methods section.

Also, in the literature review section, it might be helpful to mention which mosquito species and insecticides were evaluated in the 61 publications included for analysis.

  • This information is included for each paper in Appendix A, and a sentence has been added to the Results to refer to the variety of species and chemistries included in the sampled manuscripts.

Regarding bioassay data presentation in figures, it is a bit confusing to include the covered and uncovered treatments, given that is it a parameter that could be presented separately from other two parameters evaluated (mosquito age and number).  

  • We thank the reviewer for this suggestion which we considered carefully. We decided that the results of the covered and uncovered treatments should be included in the same figure as it was part of the same experiment, but have reworked the figure and think that the treatments are now sufficiently clear.

Given that mosquito age and number are reported as key parameters to obtain consistent results, would it be possible to perform statistical analyses that show that differences in mortality recorded are actually significant?  

  • Additional statistical analysis has been performed to determine significance.

Some minor inconsistencies in unit format used (Ex: hr. or hours?), spaces between temperature and RH values, and scientific names not italicized were noticed.

  • Corrected throughout ms.

Introduction:

Line 54: either “hr.” or “hour” should be used consistently across the text.

– Updated throughout ms

Line 110: “et al.”

– et al added and italicised

Materials and Methods:

Line 197: Appendix A not found in supplementary files

– Reattached with resubmission.

Line 208: gene names (kdr, ace-1) should be italicized

– Italicized throughout

Lines 227-228: Briefly listing other factors could be helpful.

– Details of other factors not identified for experimental testing added.

Lines 238-240: either “hr.” or “hour” should be used consistently across the text. Spaces between temperature and RH% values should be consistent across the text.

- Updated throughout ms

Table 1: check for missing periods and spaces between words.

– This is due to the table size. Table cannot be resized further to fit and improve legibility of the text. Alternatively, this table can be submitted as an additional file to allow it to be viewed in more detail.

Results:

Figure 5:  the word “days” could be deleted from the X-axis values and include in parenthesis in the X-axis label to reduce clutter.

– Updated figure

Figures 6 and 7:  may consider removing the covered (25 per test unit) treatment so that only uncovered treatments are compared. The covered vs. uncovered comparison could be just mentioned in the text, or presented in a separate graph.  Scientific names in figure legends have to be italicized.

– Figure updated. Legends updated to align with journal.

Figure 8: Scientific names in figure legends have to be italicized.

- Legends updated to align with journal.